**Peer**J

# The controlled imitation task: a new paradigm for studying self-other control

Sukhvinder S. Obhi and Jeremy Hogeveen

Social Brain, Body & Action Lab, Centre for Cognitive Neuroscience & Department of
Psychology, Wilfrid Laurier University, Waterloo, Ontario, Canada

## ABSTRACT

In the automatic imitation task (AIT) participants make a cued response during simultaneous exposure to a congruent or incongruent action made by another agent. Participants are slower to make the cued response on incongruent trials, which is thought to reflect conflict between the motor representation activated by the cue and the motor representation activated by the observed action. On incongruent trials, good performance requires the capacity to suppress the imitative action, in favor of producing the cued response. Here, we introduce a new experimental paradigm that complements the AIT, and is therefore a useful task for studying the control of self and other activated representations. In what we term the "Controlled Imitation Task (CIT)", participants are cued to make an action, but on 50% of trials, within 100 ms of this cue, an on-screen hand makes a congruent or incongruent action. If the onscreen hand moves, the participant must suppress the cued response, and instead imitate the observed action as quickly and accurately as possible. In direct contrast to the AIT, the CIT requires suppression of a self-activated motor representation, and prioritization of an imitative response. In experiment 1, we report a robust pattern of interference effects in the CIT, such that participants are slower to make the imitative response on incongruent compared to congruent trials. In experiment 2, we replicate this effect while including a non-imitative spatial-cue control condition to show that the effect is particularly robust for imitative response tendencies *per se*. Owing to the essentially opposite control requirements of the CIT versus the AIT (i.e., suppression of self-activated motor representations instead of suppression of other-activated motor representations), we propose that this new task is a potentially informative complementary paradigm to the AIT that can be used in studies of self-other control processes.

Corresponding author
Sukhvinder S. Obhi, sobhi@wlu.ca

Motor resonance, Automatic imitation Task, Self-related processing, Other-related processing,
Self-other control

## INTRODUCTION

In a popular version of the *automatic imitation* task (AIT), participants respond to a symbolic cue (usually a number) that instructs them to lift either their index finger or their middle finger. The symbolic cue is overlaid on a video showing another individual lifting their index or middle finger. When the cue and the video are incongruent, there is a reaction time (RT) cost to produce the cued movement compared to when the cue and

**How to cite this article** Obhi and Hogeveen (2013), The controlled imitation task: a new paradigm for studying self-other control. **PeerJ**
**1:e161**; DOI 10.7717/peerj.161

the video are congruent. This cost, termed "interference", is thought to be due to automatic activation of motor representations matching the observed action. Numerous control studies and conditions have been employed to demonstrate that the task really does seem to isolate automatic imitative tendencies and not simply spatial compatibility or other confounding processes (*Brass et al., 2000*; see *Heyes, 2011* for a review), though the extent to which they are etiologically similar is a matter of open debate (*Bertenthal & Scheutz, 2013*; *Cooper, Catmur & Heyes, 2012*).

In essence, the AIT assesses the degree to which an observer's motor system 'resonates' with an observed action of another individual. Successful task performance in the incongruent condition specifically involves suppressing the automatic tendency to imitate another individual, in favor of making the cued response (*Heyes, 2011*; *Santiesteban et al., 2012b*). The difference between incongruent and congruent trial RTs is robust even when participants expect the observed action to contradict their response, and thus the paradigm can be considered within the set of cognitive tasks that are generally accepted as reflecting automatic processes, such as the Stroop task (*Hogeveen & Obhi, 2013*; *Stroop, 1935*; *Tzelgov, Henik & Berger, 1992*). Notably, however, the neurocognitive bases for automatic imitation have been dissociated from Stroop interference, as neuroimaging and lesion data have shown that the two tasks are associated with activity in largely different brain regions (*Brass, Derrfuss & von Cramon, 2005*; *Brass, Zysset & von Cramon, 2001*; *Spengler, von Cramon & Brass, 2010*).

At the neural level, the automatic tendency to imitate the actions of another agent is thought to be due to mirror activity which induces motor resonance - the specific activation of motor representations that would be active if the observer themselves performed the action they are watching (*Hogeveen & Obhi, 2011*). This idea is bolstered by evidence from transcranial magnetic stimulation (TMS) studies which have consistently found an increase in motor cortical excitability when a participant watches an action, that is specific to the motor representation that would be involved in the performance of the action (e.g., *Fadiga et al., 1995*).

Over recent years the AIT, in one form or another, has been heavily employed by researchers interested in social information processing in normal participants and in participants with a range of disorders, but most notably autism spectrum disorders (ASD) (*Cook & Bird, 2012*; *Hogeveen & Obhi, 2011*; *Spengler, Bird & Brass, 2010*; *Wang, Ramsey & Hamilton, 2011*). A strong suggestion has been made that a key process involved in the AIT is the process of controlling self-related and other-related representations (*Brass, Ruby & Spengler, 2009*; *Obhi et al., 2013*; *Santiesteban et al., 2012a*; *Spengler, Bird & Brass, 2010*). Indeed, in a recent study, Santiesteban and colleagues (*2012b*) asked a group of participants to perform the opposite movement from the model onscreen for an extensive training period (counter-imitation training). The next day, they found a downstream improvement in both (i) their performance on incongruent trials of the automatic imitation task, and (ii) their ability to take someone else's visual perspective. In the perspective taking task participants were required to move objects to locations within a bookshelf as requested by another person on the other side of the bookshelf, counter-imitation training was found

to improve participants' ability to suppress their own perspective and take the perspective of the other. On the one hand, reduced interference in the AIT corresponds to suppressing influence from another agent in favor of prioritizing a more task relevant cued response. On the other hand, taking the perspective of another person involves suppressing the self-related representation, in favor of the other perspective. In this light, both tasks can be said to require control of self-related and other-related representations.

The AIT, then, is an example of a task requiring self-other control processes and in particular, the control requirement in the critical incongruent trials is to suppress the automatic imitative response, and enforce, or prioritize the cued response. In the present study, we introduce a complementary paradigm for the study of self-other control which has essentially the opposite control requirements to the AIT. We term the paradigm the "controlled imitation task" (CIT) so as to contrast it explicitly with the "automatic imitation task". In the CIT, as in the AIT, participants are exposed to a picture of a neutral hand upon which a symbolic cue in the form of a numeral is superimposed. Also akin to the AIT, the participant must lift their index finger in response to the number 1 and must lift their middle finger in response to the number 2. Critically, however, both the temporal evolution of a given trial, and the task instructions in the CIT clearly dissociate it from the AIT. With respect to trial timing, whereas the AIT cue and movement appear simultaneously, in the CIT the cue is presented *before* the onset of any observed movement. Then, within 100 ms of the cued response, the hand upon which the symbolic cue was presented makes either a congruent or an incongruent action, such that half the time, participants see an index finger lift when they are preparing an index finger lift (congruent trial), or they see a middle finger lift when they are preparing an index finger lift (incongruent trial), and vice versa for cued middle finger lifts. With respect to task instructions, whereas in the AIT participants are asked to ignore the observed movement and enforce the cued response, in the CIT if the onscreen finger moves participants are instructed to respond immediately with an identical finger movement. In essence, on incongruent trials of the CIT participants must suppress their own cued action preparation, in favor of producing the imitative response – which constitutes the opposite control requirement compared to the standard AIT. Crucially, these congruent and incongruent trials only occur on half of the total trials in a block, and in the other half of trials, the neutral hand on the screen does not move and participants simply execute the cued response. Thus, across the block, participants expect to respond to the cue, rather than simply waiting for an imperative movement to occur.

Using this paradigm, we predicted that the RT to make the controlled imitative response would be greater on incongruent trials compared to congruent trials. However, in contrast to the RT cost in the AIT, which is driven by the need to suppress imitation, the RT cost in the CIT would reflect the time required to suppress the cued response, in favor of producing an imitative response. If the congruence effects we predict are found, we suggest that this task could be a useful paradigm for the study of self-other control processes. In particular, the use of the CIT *in conjunction* with the AIT, in experiments where self-other

control is manipulated in the context of imitation (*Santiesteban et al., 2012a*; *Santiesteban et al., 2012b*) would allow researchers to be confident that what they are manipulated is indeed *control* of co-activated self- and other-generated motor representations. Beyond this, we suggest that the CIT could be a useful paradigm for the more general study of processing relating to self and other.

# EXPERIMENT 1

The aim of experiment 1 was to provide a proof of concept that an activated motor plan interferes with one's ability to imitate an incongruent observed action. To that end, participants planned to make simple index and middle finger lifts in response to the number *1* or *2*, respectively. During the response preparation period (50, 70, or 90 ms after cue-onset), an onscreen hand was most likely to remain still (baseline condition), or might have depicted an index or middle finger lift. On one subset of trials, the depicted action was congruent with the symbolic cue, and was therefore congruent with the participants' motor plan. However, on another set of trials, the depicted action was incongruent with the symbolic cue, and participants had to modify their plan to prioritize execution of an imitative response. The magnitude of the performance difference between the congruent and incongruent conditions could be informative about the degree to which a symbolically cued motor plan can interfere with the execution of an imitative action.

## Methods
### Participants
Twelve participants (9 female) between the ages of 19 and 33 ($M = 23.33$, $SD = 3.55$) completed the study. Three of the 12 participants were left-handed, and all had normal or corrected-to-normal vision. The study conformed to local ethical guidelines and informed consent was obtained from all participants.

### Stimuli & apparatus
The experiment was programmed using Superlab v.4.5 (Cedrus Corporation) and run on a Lenovo desktop computer (Lenovo Group Limited). Experimental stimuli were adapted from previous automatic imitation studies (*Brass et al., 2000*; *Cook & Bird, 2012*; *Hogeveen & Obhi, 2013*). The stimuli contained a number cue, followed by an image series depicting either an index or middle finger movement (congruent and incongruent conditions), or a still image of a hand (baseline; Fig. 1). Baseline trials made up 50% of the total trials, to ensure that participants were expecting to make the cued response and not simply waiting for the movement of the onscreen hand. The images were rotated orthogonally to the participants' hand to mitigate the influence of spatial compatibility between the observed action and the appropriate response. On baseline trials, the hand stayed in the same position for the duration of the trial (568 ms). On congruent and incongruent trials, the still hand was replaced by a number cue, which was then replaced by the first stage (34 ms), second stage (34 ms), and final stage (500 ms) of an index or middle finger lift. The final event on each trial was a blue screen, which was presented for 3000 ms to allow for delayed responses and provide participants with enough time to get situated for the next trial.

**Peer**J

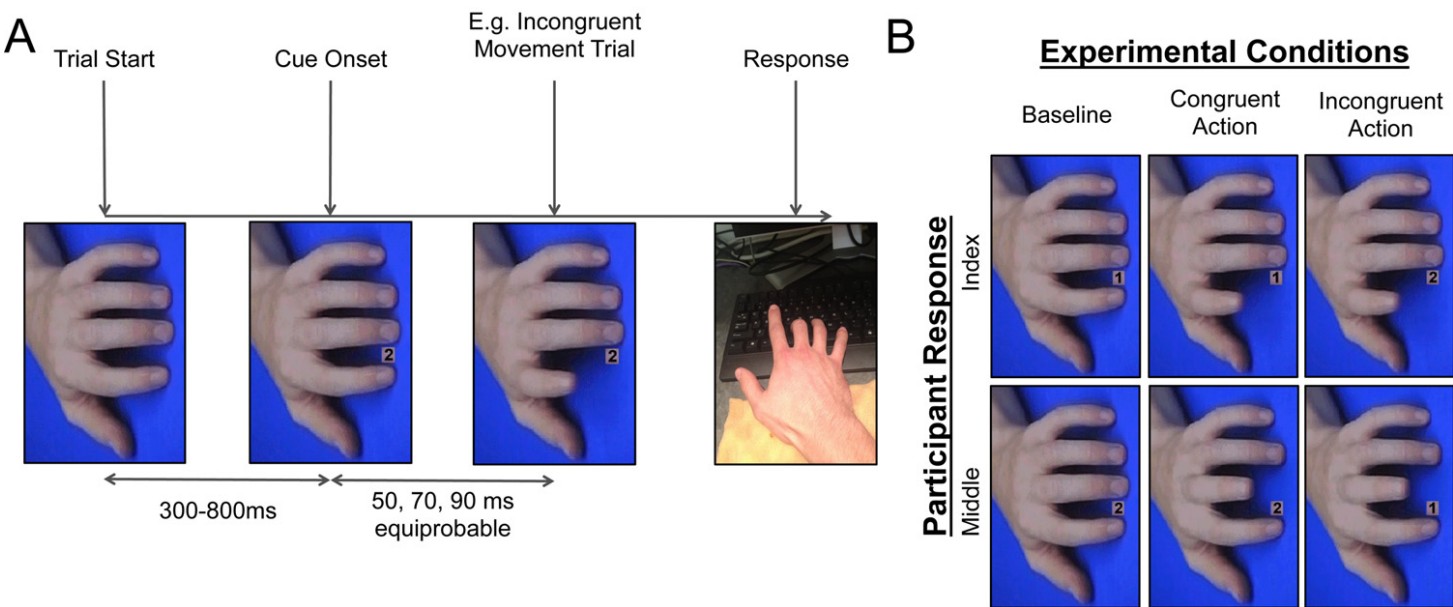

**Figure 1  Cit design.** (A) Breakdown of the events in a trial of Experiment 1, displaying an incongruent index finger movement trial as an example. (B) Breakdown of the different types of trials in Experiment 1, that were randomized within blocks. Note: the number of baseline trials was equal to the sum total of congruent *and* incongruent trials.

### *Procedure*

The experiment took place in a dimly lit cubicle. Participants were seated at the computer, told they would be holding down the *v* and *b* keys of the keyboard with their right index and middle fingers throughout the experiment, and would need to execute an index or middle finger lift as fast and accurately as possible when the number *1* or *2* was presented onscreen, respectively. Responses were made by releasing the *v* or *b* keys on a standard Lenovo keyboard. Further, participants were told: "if the hand onscreen moves, cancel your response and imitate what you see regardless of the movement you were planning to make".

The study began with a practice session containing 24 trials, which were observed by the experimenter to ensure that participants understood the task. If the participant made more than 5 mistakes, the practice section was re-run, which only occurred for one participant in the sample. The experiment proper was divided into six blocks of 32 trials, each containing four congruent index finger trials, four congruent middle finger trials, four incongruent index finger trials, four incongruent middle finger trials, eight baseline index finger trials, and eight baseline middle finger trials. Each trial began with a still image of the hand on screen (300, 400, 500, 600, 700, or 800 ms), then the response cue was presented, and after a delay period (50, 70, or 90 ms) the hand either moved or stayed still (see Stimuli & Apparatus for a description of the hand movement).

## Results

### *Data exclusion*

As described in the Methods section, the baseline condition was in place purely to ensure that participants actually prepared the cued response on each trial, and was excluded from

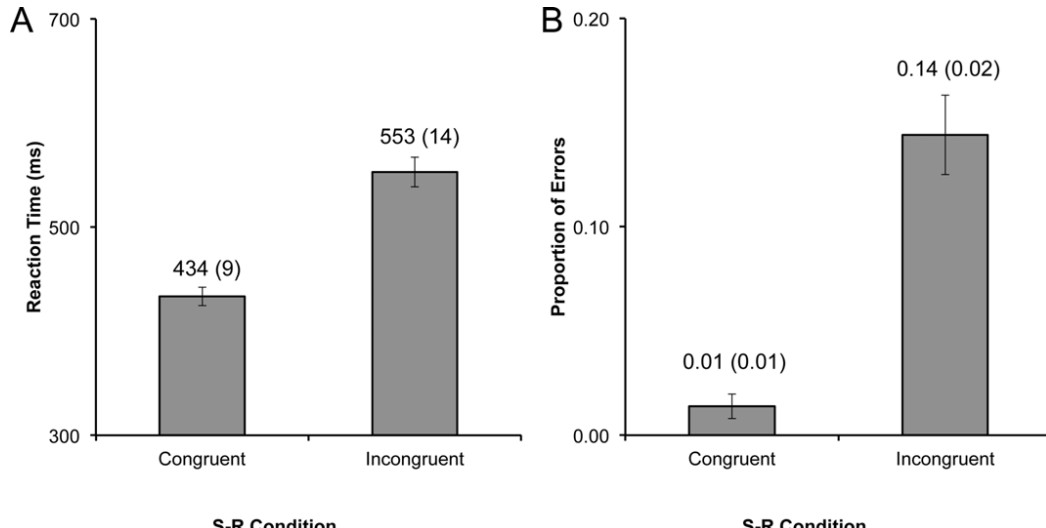

**Figure 2 Cit results.** Performance data for Experiment 1, with (A) reaction time (RT), and (B) proportion of errors (PE) plotted by experimental condition. Data labels represent the mean (and standard error of the mean) for each bar. The congruence effect was highly significant for both RT and PE (all $ps < .001$).

further analysis. Reaction time data for index and middle finger actions were collapsed and responses that were 3 SD's above or below the mean within each experimental condition (i.e., congruent, or incongruent) were excluded from statistical analysis, resulting in the removal of 0.39% of the data. The remaining congruent and incongruent trial data was then subjected to inferential statistical analysis.

### Reaction time (RT)

Each participant's average RT for correct responses was entered into a repeated-measures ANOVA with one factor (congruence: congruent, or incongruent). The effect of congruence was highly significant ($F_{1,11} = 103.06$, $p < .001$, $\eta^2 = .90$). Participants were faster on congruent trials ($M = 434$, $SE = 9$) than they were on incongruent trials ($M = 553$, $SE = 14$; $t_{11} = -10.15$, $p < .001$, $\eta^2 = .68$; Fig. 2A).

### Proportion of errors (PE)

Errors and missed responses (3.95% of total data) were summed and divided by the number of trials, yielding the proportion of errors (PE) for each participant for each condition. This data was entered into a repeated-measures ANOVA akin to the RT data, and the main effect of congruence was highly significant ($F_{1,11} = 38.11$, $p < .001$, $\eta^2 = .78$). Participants made a substantially smaller number of errors on congruent ($M = 0.01$, $SE = 0.01$) trials compared to incongruent trials ($M = 0.14$, $SE = 0.02$; $t_{11} = -6.17$, $p < .001$, $\eta^2 = .46$; Fig. 2B).

## Discussion

The RT interference effect in the well-studied AIT indexes the cost associated with inhibiting other-related processing in favor of executing a self-related task. In study 1, we establish the CIT as a potential method for measuring the cost of self-related motor

preparation to the execution of a conflicting imitative response – i.e., the cost of inhibiting self-related processing to prioritize processing of the other. Specifically, we found a robust effect in the CIT task: participants were 119 ms slower on incongruent relative to congruent trials. While this result provides reason to believe that the CIT might be a useful addition to the study of self-other related processing, it is possible that the results simply represent the cost of action modification generally, rather the cost associated with conflicting self-other representations *per se*. It is worthwhile pointing out that the size of the CIT effect (119 ms) is much greater than would be anticipated in response to a socially innocuous action modification cue (e.g. 40 ms for an auditory tone; *Obhi, Matkovich & Gilbert, 2008*), suggesting that the robust effects reported in experiment 1 were at least partially driven by the 'specialness' of the observed action. Regardless, we conducted a second experiment to assess the extent to which the CIT reflects response modification generally, or is specific to controlling co-activated self-other motor representations. To shed more light on how the CIT and AIT might be useful complementary paradigms, we also added a set of AIT blocks to experiment 2.

# EXPERIMENT 2

In experiment 2, we sought to determine whether the CIT indexes self-other control in an imitation context, or simply quantifies the cost associated with modifying a planned response. To this end, participants performed the same task as described in experiment 1, but with the addition of a moving dot condition (spatial control; *Cross et al., 2013*; Fig. 3). The dot cues were superimposed over the hand stimuli from experiment 1, and on CIT blocks participants were instructed to match what they saw onscreen whenever one of the dots, or one of the fingers, moved during their response preparation period. The difference between 'finger incongruent-congruent' interference and 'dot incongruent-congruent' interference provides an approximation of the cost associated with cancelling a self-related motor plan in favor of an *imitative* response. Admittedly, if no interaction between stimulus type and congruence were present in Experiment 2, the matching RT functions could still be driven by distinct underlying mechanisms, but such subtraction logic is often an inherent limitation of purely behavioral data (*Rumelhart & McClelland, 1986*). Experiment 2 also included a set of AIT blocks.

## Methods

### Participants

Sixteen participants (9 female) between the ages of 17 and 40 ($M = 22.44$, $SD = 5.18$) completed the study. All of the participants were right-handed, and had normal or corrected-to-normal vision. The study conformed to local ethical guidelines and informed consent was obtained from all participants.

### Stimuli & apparatus

Experimental stimuli & apparatus were similar to experiment 1, with two exceptions: (i) black dots were superimposed over the fingernails of each finger, and (ii) on trials where the finger or dot moved, the movement occurred at once, rather than containing

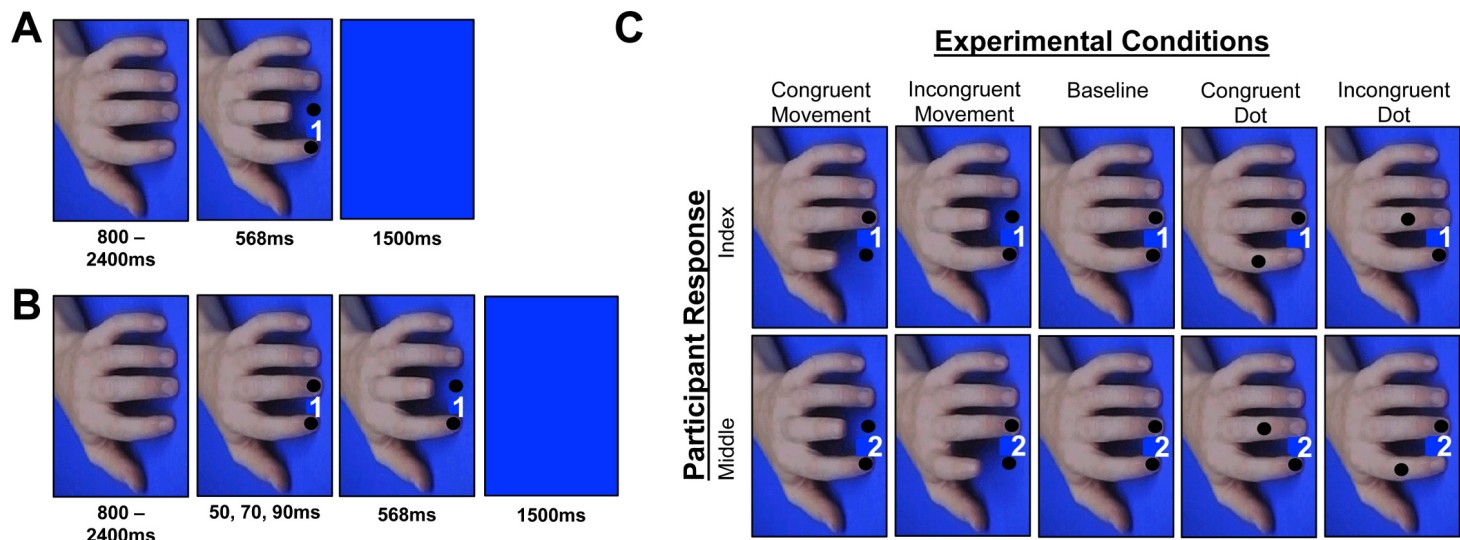

**Figure 3 Ait and cit design experiment 2.** Breakdown of the trial events and conditions from experiment 2. (A) AIT displayed the cue and finger or dot movement simultaneously, whereas (B) the CIT displays the cue, followed by a finger or dot movement that appears during the response preparation period. (C) All of the conditions that took place in the experiment, which were randomized within both the AIT and the CIT blocks. Note: the cue numbers have been exaggerated for display purposes only. Also, whereas Experiment 1 used picture sequences, Experiment 2 immediately moved to the final state of the action, producing an apparent motion effect that has been shown to elicit robust effects in past AIT studies (*Catmur & Heyes, 2011*; *Press et al., 2005*).

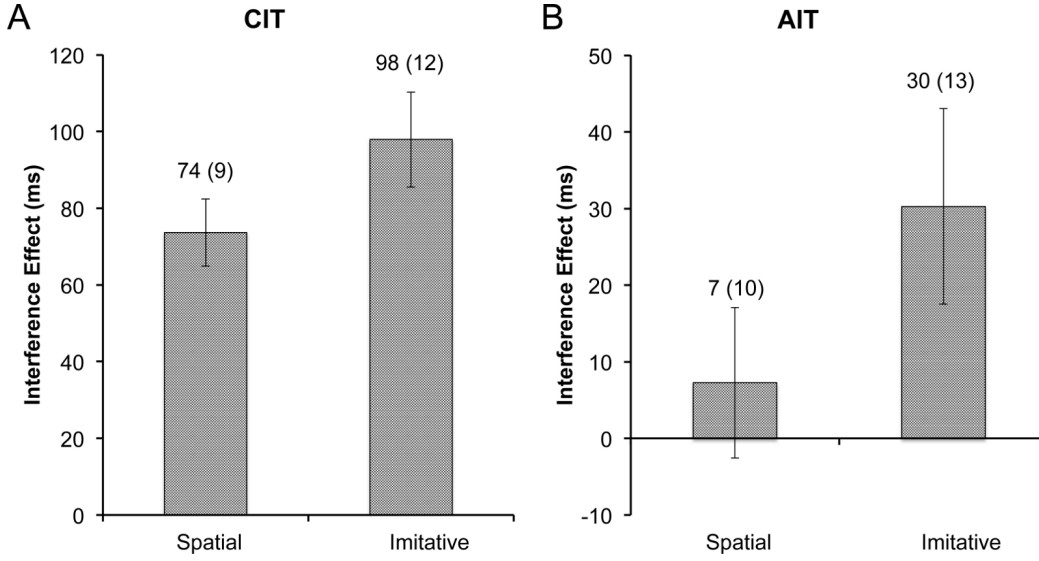

**Figure 4 Experiment 2 interference.** Size of the RT interference effect in experiment 2, with (A) the CIT and (B) the AIT interference effects plotted by cue type. Data labels represent the mean (and standard error of the mean) for each bar. The interference effect was significantly higher for the imitative cue in both the CIT and the AIT tasks ($ps < .05$).

intermediate movement stages (cf. *Catmur & Heyes, 2011*; *Press et al., 2005*) (Fig. 3). On trials where the fingers moved, the dots remained in position; whereas on trials where the dots moved the fingers remained in position. The final locations of the dots in the spatial control conditions were positioned at the final locations of the finger movements in the imitative conditions.

### Procedure

The CIT and AIT each contained five experimental conditions randomized within blocks: congruent dot, incongruent dot, congruent finger, incongruent finger, and baseline (Fig. 3). The trial events in the CIT were nearly identical to experiment 1, except experiment 2 contained a longer neutral stimulus and a shorter blue screen. Participants completed a practice CIT block (32 trials: 4 congruent dot, 4 incongruent dot, 4 congruent finger, 4 incongruent finger, 16 baseline) followed by experimental CIT blocks (4 blocks × 48 trials) that contained a total of 24 trials per operant (i.e., congruent, incongruent) condition. Akin to experiment 1, on the CIT blocks participants were instructed: "if the hand or the dot onscreen moves, cancel your response and match what you see regardless of the movement you were planning to make". Participants who made >5 errors during the practice block in the CIT ($n = 4$) performed the practice twice.

In experiment 2, participants completed a practice AIT block and four experimental AIT blocks containing the same randomized trial type distributions as the CIT. The order of presentation for the two tasks was counterbalanced across the sample. Prior to the AIT, participants were instructed to perform index and middle finger lifts in response to the numbers *1* or *2*, respectively, and to execute that plan "no matter what the hand or dot onscreen does". Thus, overall the stimuli and task demands required for the AIT and CIT were identical, except in the former enforcing self was required to inhibit other, whereas in the latter enforcing other was required to imitate and cancel a planned response.

## Results

### Data exclusion

As in experiment 1, the baseline condition was in place to ensure that participants expected to perform the cued response on the majority of trials, and was excluded from further analysis. Within-subjects data exclusion was identical to the criterion set in experiment 1, resulting in the removal of 0.33% of all data. Further, a between-subjects criterion of 3 SD away from the mean was violated by one participant's mean RT in the congruent finger condition, and this participant was removed from the dataset (leaving $n = 15$). It is worthwhile to note that the subsequently reported inferential results were identical with or without the outlying participant.

### CIT and AIT RT analyses

Mean correct response times on the CIT blocks were entered into a 2 (stimulus type: dot, finger) × 2 (congruence: dot, finger) repeated measures ANOVA (rmANOVA). There was no effect of stimulus type ($p = .14$), but the effect of congruence was highly significant

($F_{1,14} = 80.95$, $p < .001$, $\eta^2 = .85$). Most importantly, the CIT RT analysis yielded a significant interaction between stimulus type and congruence ($F_{1,14} = 6.00$, $p < .05$, $\eta^2 = .30$). To elucidate the interaction, interference effects were calculated for the dot and finger conditions, which were then entered into a rmANOVA with one factor (stimulus type: dot, finger). The mean interference effect for the spatially-matched dot control ($M = 74$, $SD = 34$) was significantly smaller than the interference effect for the imitation condition ($M = 98$, $SD = 48$; $F_{1,14} = 5.93$, $p < .05$, $\eta^2 = .30$; Fig. 4A).

Next, mean correct RTs on the AIT blocks were entered into a 2 (stimulus type: dot, finger) × 2 (congruence: dot, finger) rmANOVA. Neither of the main effects were significant ($ps > .1$), but importantly the interaction between stimulus type and congruence was significant on the AIT ($F_{1,14} = 8.21$, $p < .05$, $\eta^2 = .37$). Akin to the CIT analysis, the interference effects on the AIT were computed and entered into a rmANOVA with one factor (stimulus type: dot, finger). Again, akin to the CIT, the dot control ($M = 7$, $SD = 38$) had a much smaller effect than the imitative condition ($M = 30$, $SD = 49$; $F_{1,14} = 8.23$, $p < .05$, $\eta^2 = .37$; Fig. 4B) in the AIT context.

The lack of a main effect of congruence in the AIT was surprising, given the fact that previous AIT studies have found a main effect of congruence even when a similar spatially-matched control condition was included (*Cook & Bird, 2011*; *Cook & Bird, 2012*). One possible reason for this discrepancy could be that the order of the two tasks (i.e., CIT then AIT vs. AIT then CIT) confounded congruence in the AIT. To isolate any order effects, the 2 × 2 rmANOVAs described for the CIT and AIT were re-run as analyses of covariance (rmANCOVA), regressing out the task order term. Interestingly, the rmANCOVAs for the CIT ($p = .19$), and the AIT ($p = .56$), no longer contained significant interaction effects. Participants who performed the AIT before the CIT showed a significant interaction on the CIT, with dots ($M = 78$, $SD = 28$) having a smaller interference effect than fingers ($M = 110$, $SD = 54$; $F_{1,6} = 6.06$, $p < .05$, $\eta^2 = .50$), whereas participants who performed the CIT immediately did not show a stimulus type preference ($p = .29$). Conversely, participants who performed the CIT first showed a significant stimulus type preference on the AIT, with dots ($M = 17$, $SD = 46$) having a smaller effect than fingers ($M = 42$, $SD = 53$; $F_{1,7} = 6.39$, $p < .05$, $\eta^2 = .48$), and participants who performed the AIT immediately showing no difference as a function of stimulus type ($p = .18$).

## Discussion

After documenting a large CIT interference effect in experiment 1, we conducted experiment 2 in order to replicate the CIT, and to differentiate controlled imitation from response modification by including a non-social cue condition. The results from experiment 2 demonstrate again that the size of the RT cost associated with cancelling a self-related movement in favor of imitation is quite large (98 ms), and that it is greater than the cost in a well-matched spatial-cue condition (dots: 74 ms). The finding of a greater interference effect to the finger stimulus in both the CIT and the AIT is likely due to the greater degree of overlap between the conflicting representations relative to the moving dot conditions. Whereas the spatial location and movement affordances are matched

between the finger and dot stimuli, the former contains a topographically congruent or incongruent movement by a model that is not present in the latter. Without saying anything about their ontogeny (*Cooper, Catmur & Heyes, 2012*), the present finding is consistent with suggestions that, once developed, imitative compatibility reflects a form of stimulus-response compatibility that is unique relative to non-imitative spatial-cue-driven effects (*Catmur & Heyes, 2011*). Further, we suggest that experiment 2 utilizes a useful control condition for the CIT, which future studies attempting to manipulate imitative self-other control should include to differentiate controlled imitation from general response modification processes.

## GENERAL DISCUSSION

In two experiments, we report a novel task requiring the online control of competing self- and other-generated motor representations. We have called this task the controlled imitation task (CIT), and suggest that it is a useful complementary task to the well-studied AIT. On each experimental trial, participants had to prepare a cued response, but on 50% of trials they were required to suppress this response and instead prioritize the production of an imitative response that was either congruent or incongruent with the cue. The robust result is that participants were significantly better at imitating an observed movement when it matched the response cue, relative to when it did not match the cue (experiment 1), and that this effect was reliable and of a much larger magnitude than the traditional AIT (experiment 2).

We suggest the CIT provides a valuable method for examining the influence of the *self* with respect to self-other control in the imitative context. To the extent that the production of a symbolically-cued response reflects a task specific self-activated process, the incongruent condition in the CIT can be thought of as requiring suppression of this process in favor of producing an imitative response. Thus, the present results nicely complement typical AIT data: in the AIT interference effects reflect the cost of inhibiting an other-activated motor representation in favor of a cued response, whereas in the CIT interference effects reflect the cost of inhibiting a self-activated motor representation in favor of imitation.

Experiment 2 enables us to suggest not only that controlled and automatic imitation are more pronounced than general response modification and inhibition, respectively, but also that this biological preference is driven by recent experience. Typically, biological-specificity effects are present a priori (*Kanwisher, McDermott & Chun, 1997*; *Kilner, Paulignan & Blakemore, 2003*), but here preference for the imitative stimulus varied as a function of task order. Participants who engaged in the CIT initially, showed a finger-preference on the subsequent AIT, and participants who did the AIT first showed this preference on the subsequent CIT, but neither group showed a stimulus preference during the initial task. This suggests that inhibiting imitation at time 1 (AIT) makes it more difficult to perform imitation at time 2 (CIT), which was not true for the dot control condition. Conversely, performing imitation at time 1 (CIT) makes it harder to inhibit imitation at time 2 (AIT), which again was not the case for the dot control condition.

Thus, it appears the shared self-other representational system that is invoked to explain the traditional AIT is also engaged by the CIT. Furthermore, this system seems to be strongly affected by recent experience (*Catmur, Walsh & Heyes, 2007*; *Heyes et al., 2005*; *Hogeveen & Obhi, 2012*).

Beyond the relationship between the CIT and AIT, there are several interesting issues and possibilities for future work that could be studied with the CIT. For example, for studies designed to manipulate self-other control, the CIT provides an excellent and quite well matched experimental paradigm with which to contrast AIT, and investigate task-appropriate shifts in processing towards self or other in the motor domain. A recent study by Santiesteban and colleagues (*2012b*) found that imitation-inhibition training on day 1 suppresses the automatic tendency to imitate on day 2, in order to facilitate task performance which necessitates production of the cued response. Whereas the size of the AIT interference effect was reduced after training, accuracy on a visual perspective-taking task was improved, suggesting that processing of the other person was enhanced or inhibited in a task-appropriate manner (*Santiesteban et al., 2012b*). Despite the impressive nature of this finding, the CIT provides a closer other-centric homologue for AIT than visual perspective taking. Thus, if counter-imitation training has a functional impact on *task dependent* control of self- and other-generated *motor* representations, we would expect to observe an increased ability to suppress the self-activated motor representation in the CIT. Seeing a pattern of results that is consistent with suppression of self-related activity in the CIT combined with suppression of other-related activity in the AIT would bolster the notion that the counter-imitation training improves task specific control of self-other processing.

In addition to comparing the AIT and CIT directly after a self-other control manipulation, the extant difference between the two tasks could be used to index individual differences in self-related bias. On their own, the size of the CIT or AIT interference effects may reflect self-other distinction, control, or both. However, by subtracting the AIT from the CIT, experimenters could quantify the extent to which self-other control is biased in favor of the self, which could then be related to personality variables hypothesized to relate to self-other control. It remains for future work to investigate this suggestion further.

In the non-social context, another very interesting aspect of the CIT is that it can be clearly contrasted with the *truncation paradigm*, in which internal generation of action is pitted against external cueing of action in non-imitative contexts (*Obhi & Haggard, 2004*; *Obhi, Matkovich & Chen, 2009*; *Obhi, Matkovich & Gilbert, 2008*). In particular, in the truncation paradigm participants begin the trial preparing to make an index finger movement at a time of their own choosing, but are interrupted on a portion of trials by a tone prompting production of the same movement. That is, in the truncation paradigm, participants prepare to make a self-paced action and are interrupted with a non-social external cue to produce the same action or a different action, whereas in the current study the external cue to action is a movement by another person. Given the dissociation between the ability to inhibit imitation in the AIT and the inhibition of other overlearned responses (*Brass, Derrfuss & von Cramon, 2005*; *Spengler, von Cramon & Brass, 2010*),

it would be interesting to determine whether the CIT and action modification in the truncation paradigm depend on similar or specialized neural substrates. Implementation of this line of enquiry would necessitate modification to the CIT to include a self-paced action, as opposed to a cued action. Such experiments could be useful for understanding the domain-generality or domain-specificity of the functional mechanisms underlying self-other control.

In summary, in two experiments we introduce a new paradigm that we have termed the "controlled imitation task". Using an identical stimulus set to the established AIT, the CIT yields robust interference and facilitation effects that appear to shed light on self-other control processes, and that specifically tap into the capacity to suppress, or harness self-activated motor representations to facilitate fast production of an other-activated response (cf. *Brass, Ruby & Spengler, 2009*). We have outlined just two possible areas of inquiry where the CIT could be a useful paradigm; self-other control, and the more general exploration of internally versus externally triggered action generation. It remains for future work to consider exactly how this paradigm can be employed to further understanding in these areas of study.

### Funding
This work was supported by a standard research grant from the Social Sciences & Humanities Research Council of Canada: 410-2011-2103. The funders had no role in study design, data collection and analysis, decision to publish, or preparation of the manuscript.

### Grant Disclosures
The following grant information was disclosed by the authors:
Social Sciences & Humanities Research Council of Canada: 410-2011-2103.

### Competing Interests
The authors declare there are no competing interests.

### Author Contributions
- Sukhvinder S. Obhi conceived and designed the experiments, analyzed the data, wrote the paper.
- Jeremy Hogeveen helped conceive and design experiment 2, performed the experiments, analyzed the data, wrote the paper.

### Human Ethics
The following information was supplied relating to ethical approvals (i.e., approving body and any reference numbers):
Wilfrid Laurier Ethics Approval REB code: 3627.

Peer**J** _______________________

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
