# Peer review of "The controlled imitation task: a new paradigm for studying self-other control"

_PeerJ, doi:10.7717/peerj.161_

## Round 0.1 · original submission · Major Revisions

The reviewers made a number of important comments that you may want to address in your revision. I also read the paper and have some comments, which partially overlap the comments of the reviewers.

Your paper reports a clever new experimental variation of the automatic imitation task (AIT). You propose that this new task, the controlled imitation task (CIT), can be used in studies “of self-other control processes and more generally, internally generated versus externally triggered actions.”

I think that with regard to the self-other control processes the task lacks an important control condition, that is, non biological cues requiring subjects to override their initially planned action. In this control condition, rather than seeing an action 100 ms after the cue had given subjects instructions on which action to perform, subjects would see either another symbolic or a spatial cue. The RT costs in the incongruent trials of this control task, subtracted to the RT costs in the CIT, would give us a true estimate of self-other control process. Without this control task, the only thing we can conclude is that the whopping RT costs (120 ms!) shown in this study are costs in reprogramming an action. Indeed, I am not even sure one can conclude that the task in its current form can give us an index of “internally generated versus externally triggered actions”, since the first planned action is also externally cued.

Two other minor comments:

Why 100 ms? What happens if you manipulate that time interval? Does the RT cost get bigger, smaller, it disappear completely? These are definitely interesting follow up experimental manipulations.

There are documented orthogonal spatial s-r effects. I think you should look into that literature and make sure that those effects do no play a role here.

·

Basic reporting

I find the manuscript to be well-written, clear, and concise. I enjoyed reading it. The reporting is simple and straight-forward.

Experimental design

The experiment as it stands appears to be well-executed, but I do feel that the experiment is lacking an important control which would allow the authors to draw conclusions that they draw about the usefulness of this task for studying self-other issues. In order to say this result has anything to do with imitation, it would be important to use a cue that was not another person's movement. The current results, which show that people were slower to respond when their instruction was changed mid-trial by a hand moving, may be explained rather simply without invoking any explanation relating to imitation: people make slower responses when their instruction is changed mid-movement compared to when they are not. In fact, it’s hard to see how any other result would be obtained regardless of what the interrupting cue was. It is clearly going to take more time to change a response than it is to not change a response.

Therefore, to make the case that this task has any relevance to imitation, the authors would need to show that the imitative cue itself has some special power to interfere with self-inhibition, over and above the simple cost of changing action mid-stream. This concept is briefly discussed in the discussion, but it seems to me that it is crucial to the interpretation that the task involves a conflict between self and other representations, which is the way the manuscript frames the usefulness of this task. It would be a fairly simple control condition to add to the experiment.

(
-The N is fairly small for a simple behavioral study, although the results are significant nevertheless. I wonder about the impact of the handedness of the participants, given that 3 of the 12 were left handed. It appears that everyone used the right hand to respond and the cues from the figure appear to be a left hand such that participants were doing "mirror imitation".

-How is reaction time measured? Is it the time when the key on the keyboard is released?
)

Validity of the findings

As discussed above, it seems to me that the results don't directly speak to the issue of imitation or self/other representation, and instead may be explained in simple motor terms: changing a target response midstream leads to slower responses compared with not changing it. I don't think the manuscript succeeds in making the case that this novel task is measuring anything about representations of self-other control as promised in the title.

That said, I like the idea of this task, and if it turns out that the interference effect is related to representations of observed movement it has the potential to be very useful as a complement to the AIT.

Additional comments

No comments.

Reviewer 2 ·

Basic reporting

The authors introduce a novel task, based on the automatic imitation task, which aims at testing for prioritization of imitative responses. The basic rationale of the task is a useful addition, and if constructed well is likely to spawn a large number of studies using this addition, or extensions thereof.
- It would be helpful to elaborate in more detail why the effects in the AIT are not explained by other processes, such as preparation and consequent competition resolution. In other words, what is so special about this specific process, or is there a unifying account – intuitively, having dedicated but separated systems for imitation and selection seems odd. For example, to which degree can prior expectation explain the reviewed results – participants have to respond against an expectation? It is hardly surprising that action representations with higher prior probability will be faster to express than unlikely ones, irrespective of the type of information on which such an expectation is build? See also my point below regarding the CIT. I also note that I am not an expert on action observation or imitation, so the difference may be obvious and thus easy to point out.
- As far as I understand, the CIT is essentially a countermanding task, with matched probabilities. Other cousins include reprogramming tasks (cf work by Rushworth) and stopping tasks. I can see how the AIT is incomplete, and the CIT is therefore a useful addition, but embedding the task into the vast literature on related inhibition tasks would seem useful. This point is essential as it concerns the construct validity of the task, and this reviewer is not entirely sure the task divorces itself from the various paradigms in this literature.
- It seems to me that the authors basically aim to extend the AIT to become a fully factorial task. Instead of being complementary, this author wonders if they are indeed both necessary, i.e. accurate inference requires both to be tested. One may be tempted to attribute any outcome to ‘other’ vs ‘self’ generated movements, but this essentially requires comparison of both, and neither task achieves that alone. Viewed in this way, the authors basically add another factor to the AIT, which distinguishes reference frame. That may or may not be seen as a novel task, but in any case advocating the use of the CIT as standalone seems to suffer from the same limitations as using the AIT alone, i.e. poor control for “reference frame”.
- Then again, having said the above, comparability between the AIT and CIT is somewhat limited because the temporal structure is different. One may deem this unlikely to explain any observed differences but in terms of experimental design and control this seems to be a problem. It would be useful to device a novel task in which the temporal structure in both tasks is matched
- Please report measures of effect size
- tDCS is a terrible method for anatomical inference. Such suggestions should be avoided as they permeate the field in unhealthy and incorrect ways. More generally, anatomical predictions would seem only the second step. First, what about the novel hypotheses concerning processing of observed actions that can now be addressed? As per my comment above, it would seem also that any anatomical prediction would have to include the extensive work on conflict, inhibition etc, and this reviewer wonders whether the current task would reveal anything surprising here.
- Related, if the authors want to advocate the use of this task, it may be useful to anticipate modifications. One could vary the likelihood of incongruent/congruent trials, for example, which may help to address the previous issue about preparation/prior expectation. The authors use a fixed interval of 100ms between cue and target stimulus, which may not be optimal (nb. How was this interval chosen?). In any case, it may be useful to explore the temporal nature of the process, by varying this interval similar to SSRT tasks. Note also that there is debate about the nature of these processes, e.g. whether the selection and stop process occur in parallel or not, and what the dynamics of this process are.

Experimental design

see above

Validity of the findings

see above

Additional comments

see above

·

Basic reporting

No comments

Experimental design

No comments

Validity of the findings

No comments

Additional comments

In this paper, the authors present a novel task that requires participants to suppress self-activated motor reponses following an initial cue. The authors argue that this task, which complement the well known « automatic imitation task », may be used to study self-other representations.

The paper presents a good argument for the development of such a task, and the protocol and task appear well thought. I do not have concerns regarding the manuscript as it stands. I wonder, however, how performance on the CIT correlates with performance on the classical AIT… It would have been quite interesting to have participants perform both tasks…

---

## Round 0.2 · accepted · Accept

The addition of Experiment 2 not only clarifies the nature of the results in Experiment 1 but also provides strong evidence of how temporal dynamics are powerful modulators of these effects. An excellent study that will undoubtedly inspire more research on this important topic.